# Acoustic Beam Characterization and Selection for Optimized Underwater Communication

**Akram Ahmed *** and **Mohamed Younis**

Dept. of Computer Science and Electrical Engineering, University of Maryland Baltimore County, Baltimore, MD 21250, USA
* Correspondence: akrama1@umbc.edu; Tel.:+1-410-455-8794

**Abstract:** To increase underwater acoustic signal detectability and conserve energy, nodes leverage directional transmissions. In addition, nodes operate in a three-dimensional (3D) environment that is categorized as inhomogeneous where a propagating signal changes its direction based on the observed sound speed profile (SSP). Coupling 3D directional transmission with frequent node drifts and the varying underwater SSP complicates the process of selecting suitable transmission angles to maintain underwater communication links. Fundamentally, utilizing directional transmission while nodes are drifting causes breaks in established communication links and thus nodes need to find new angles to reestablish these links. Moreover, selecting arbitrary transmission angles may lead to overlapping beams or result in leaving an underwater region uncovered. To tackle the abovementioned challenges, this paper proposes an autonomous beam selection approach that optimizes underwater communication by selecting non-overlapping beams while mitigating the possibility of missing a region, i.e., maximize coverage. Such optimization is achieved by utilizing a structured angle selection mechanism that accounts for the capability of the used transducer. Moreover, we introduce an algorithm suited for resource constrained nodes to classify rays into different types. Then we divide the underwater medium into regions where each region is identified by the limits of the coverage area of each ray type. Finally, we utilize the limits of these regions to aid nodes in selecting the best ray to reestablish communication with drifted nodes. We validate our contribution through simulation where actual SSPs are leveraged to validate the beam classification process.

**Keywords:** underwater beam selection; underwater acoustic; underwater inhomogeneity

---

## 1. Introduction

Recent advances in acoustic underwater networks (AUN) have enabled the development of applications such as environmental state monitoring, search and rescue, seabed profiling, and coastal surveillance [1]. In an AUN, a mix of mobile, tethered and freely floating nodes must establish and maintain communication links in an ad-hoc manner while ensuring certain quality of service [2]. Furthermore, forming and maintaining a connected network topology for these applications proves beneficial where nodes must collect and exchange data to accomplish their mission and to set new objectives. Surveying the area of a crashed airplane is an example where nodes in an AUN must collaborate to enable localization and retrieval of the Blackbox [3]. According to Pranitha and Anjaneyulu, acoustic signals are deemed the best communication means among nodes that are more than 100 m apart due to their low attenuation and absorption in underwater environments [4]. Fundamentally, as the frequency of the transmitted signal increases, the underwater attenuation and absorption losses rapidly grow. On the other hand, utilizing frequencies less than five kilohertz to overcome these losses degrades the ability of achieving high data rates among communicating pairs.

Thus, frequencies in the range of tens of kilohertz are favored as they strike a balance between achievable bitrate and the encountered underwater losses. Moreover, directional transmissions, rather than omni, are preferred since they overcome underwater attenuation and consume less energy. However, surface and bottom reflections, high propagation delays, drifting of underwater nodes and refraction of acoustic signals still pose major challenges to the design and operation of AUNs.

Underwater Propagation of Acoustic Signals: The underwater environment is characterized as an inhomogeneous medium, such inhomogeneity is due to point-to-point changes in underwater characteristics such as temperature, salinity, and pressure (depth) [5]. Essentially, underwater inhomogeneity is mainly manifested by means of observing the sound speed profile (SSP) of the water column. Such variation of sound speeds (SS) along the propagation path of acoustic signals causes refraction and often yields a continuous change in the gazing angle of a propagating signal. Thus, to find the propagation path of transmitted acoustics signals, ray tracing mechanisms are leveraged to infer the trajectory of each ray. Although, current ray tracing algorithms yield a good approximation of the actual acoustic signals, they require estimating a propagation path within each layer and then concatenating them to come up with the full path. Moreover, maintaining communication with mobile or drifted nodes often requires estimation of the best communication angle that is usually not necessarily contiguous to the previous one. To illustrate, let us consider the example in Figure 1a, where initially node $n_{R1}$ lies on the path of the ray transmitted by node $n_T$ at an angle $\phi_T = 12°$. As node $n_{R1}$ drifts horizontally to the right, $n_T$ shifts to the ray transmitted at angle $\phi_T = 10°$ to maintain communication. When $n_{R1}$ drifts further to the right, transmitting at smaller angles will not establish communication with $n_{R1}$ since all these rays will refract until a local extremum point (LEP) is disclosed along the propagation path prohibiting the ray from reaching $n_{R1}$, as illustrated in Figure 1a. Thus, node $n_T$ must use a different type of ray to reach $n_{R1}$ where a ray with a single LEP is to be used to reach $n_{R1}$. Therefore, it is not enough for nodes to know the location of neighboring nodes, but they also need to classify rays into different types and determine the ranges of each one to enable switching among rays to maintain a communication link. Figure 1a indicates that when SSP conditions are favorable to produce multiple differentiable LEPs, the angle of transmission plays an important role in determining the location of the LEP. Specifically, as the angle of transmission changes from 0 to 10°, we observe that horizontal range to a LEP grows, indicating that for a transmission angle more than 4°, only few iterations are needed to determine the number of LEPs within the transmission range of a node. Moreover, for transmission angles less than 4°, a node may resort to a straight-line model as the variation in angle is minimal.

Beam Selection Challenge: To establish an underwater communication link, node $n_T$ typically makes an omni-directional transmission to reach a certain receiver node $n_R$. Node $n_R$ often detects multiple copies of the transmitted acoustic signal by $n_T$ due to the multipath effect where the path of each received signal is further aggravated by refraction experienced due to SS variations along its propagation path. Such multipath effect suggests that when directional transmission is used, a communication link to a neighbor can be established using multiple angles. Moreover, in shallow water scenarios, refraction along the path is often ignored and the multipath signals are classified into four types, namely direct, surface reflected, bottom reflected and surface bottom reflected rays [6]. On the other hand, the deep-water multipath signals are quite different where losses experienced by the surface bottom reflected rays are often high; making these signals nondetectable due to the large separation between both surfaces. Moreover, when the acoustic signal moves from a lower SS region to a higher one, refraction may cause a propagation directional reversal along the depth generating a LEP along the propagation path, as illustrated in Figure 1a. As shown in Figure 1b, different ray types can be established among communicating pairs in deep-water setups where a transmitted beam may cover more than one type. Moreover, selecting two overlapping beams may result in utilizing the same ray type in establishing the communication link and thus wasting the energy of the transmitter. Furthermore, selecting disjoint beams may result in missing the ray types connecting the pairs and thus placing the neighbor in the shadow zone. Given these challenges and the fact that nodes in AUNs

are usually sparsely located [3], a method is required to govern the selection of transmitted beams to ensure minimal overlaps while avoiding shadow zones, as discussed next.

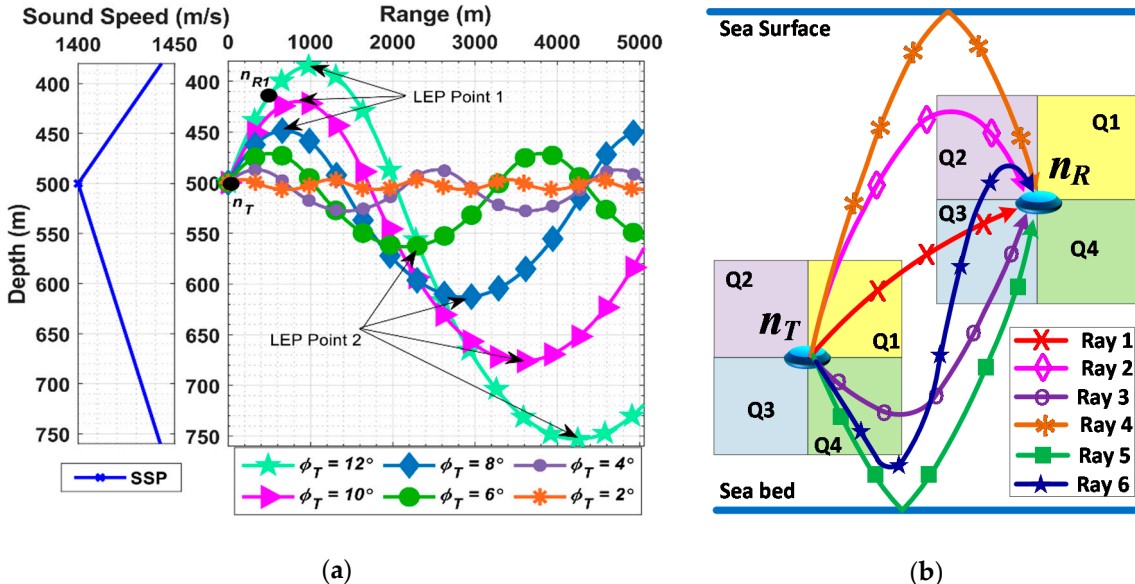

**Figure 1.** Different types of underwater ray configurations based on their propagation path. Part (**a**) illustrates the simulation results of Bellhop ray tracing algorithm for transmitted signals at small angles in the extreme SSP gradients shown on the left, while part (**b**) shows (i) a direct ray (Ray 1), (ii) rays with single Local Extremum Point (LEP) (Rays 2 and 3), (iii) reflected rays (Rays 4 and 5), and (iv) multi LEP rays (Ray 6).

Shadow zones occur in two different conditions. The first condition is encountered when nodes select transmission angles that form disjoint beams, i.e., the angle between neighboring beam edges is greater than zero. An example of such shadow zone is shown in Figure 2, where the Bellhop ray tracing algorithm is utilized to determine the propagation path of selected beams [7]. In Figure 2, when using beams 3 and 4, nodes located in the dotted region will not hear the message sent by the transmitter. Such beam selection results in generating a shadow zone in the gaps observed between beams. The second shadow zone condition is related to the underwater losses where the losses are fundamentally due to the absorption and spreading of acoustic wave where spreading dominates in long-ranges [8]. Since each transmitted acoustic beam experiences unique refraction conditions, the distance at which the shadow zone is created is dependent on the spreading pattern. To illustrate this fact, consider beam 4 in Figure 2 where the loss rate rapidly increases due to the divergence of beam edges. Thus, when the spreading reaches a point where the signal power drops below the receiver sensitivity, a shadow zone is created. Since nodes residing in the shadow zones cannot detect transmissions sent by neighboring nodes, avoiding shadow zones for nodes becomes inevitable to form a connected network.

On the other hand, the overlapping beams can be categorized into two types. The first, shown as the striped region (Zone 1) in Figure 2, is realized when selecting adjacent beams with their center angles deviation less than their beam widths. A beam of the second type, illustrated as the checkboard area (Zone 2) in Figure 2, is formed due to reversal of propagation direction along the depth. In principal, the second type is not experienced unless either a surface reflection takes place, or the propagation path experiences a LEP. Hence, for a receiver in an overlapping zone, multiple transmissions from the same sender are received which either result in wasting the transmitter's energy or increasing the probability of collision. Therefore, the selection of transmission angles is critical for avoiding these problems.

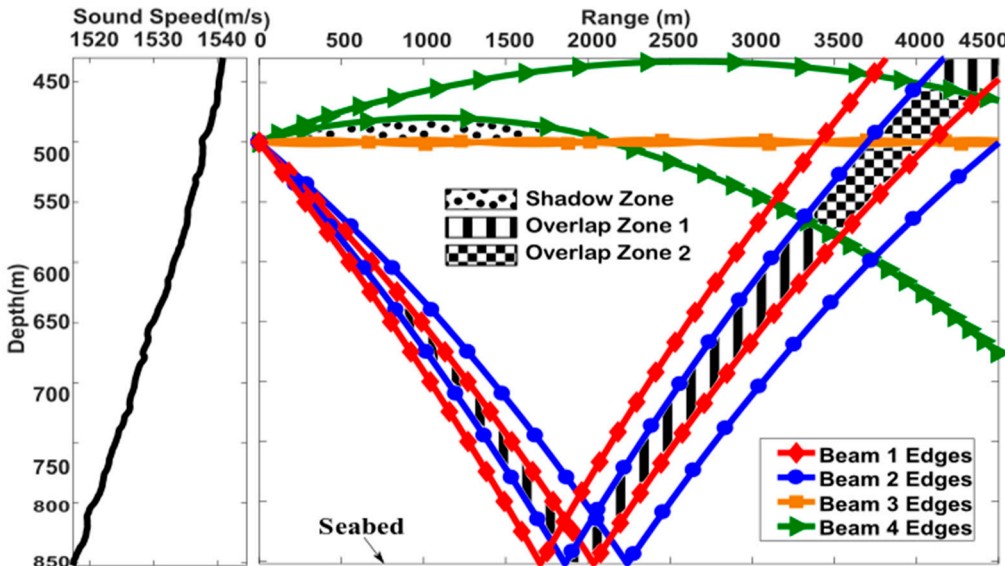

**Figure 2.** Illustration of shadow and overlapping regions where a transmitter is placed at depth of 500 m while the seabed is at 850 m. The left figure shows the sound speed profile while the right one shows the ray propagation path where the inhomogeneity of the medium complicates the spreading pattern of waves. Nodes laying in the shadow zones will not detect the transmitted waves while node in the overlapping regions receive multiple transmissions.

Summary of Contribution: To tackle the challenges presented above, this paper proposes a beam selection mechanism for deep-water setups that aids node in utilizing the directional transmission capability and avoiding generation of both overlapping beams and shadow zones in a three-Dimensional (3D) environment. Essentially, we leverage a geodesic grid to define a hypothetical 3D structure around the node and then utilize it to determine transmission angles that minimize beam overlapping while avoiding shadow zones. Moreover, since most of the ray tracing algorithms are designed to work in two-dimensional subspace (2D), we propose a projection scheme that maps the 3D underwater environment into an equivalent 2D where the SSP structure along the depth is retained, i.e., preserve the inhomogeneous effect of the environment on the traversing acoustic rays. Moreover, since most legacy systems have limited computational capabilities, such 2D mapping can be used to lower the computational burden on nodes while ensuring accuracy. Since knowing the last location of a drifted neighbor is not enough to determine the best angle to reestablish communication, we propose a beam classification technique that aids nodes in determining the most suitable angles to cover different geographical regions. Specifically, we utilize the 2D environment and the known SSP of the water column to categorize rays into three distinct types, namely direct rays, reflected rays and refracted rays experiencing LEPs. Then based on the expected coverage area of each ray type, we divide the underwater environment into four distinct geographical regions. We then use the boundaries of these regions to suggest ray types that increases the chances of reestablishing communication with drifted nodes.

The paper is organized as follows. The related work is summarized in Section 2. Section 3 discusses the system model and covers basic concepts regarding the used transducer, the layering and SSP estimation mechanism, surface-caused signal reflection and ray tracing. Section 4 goes over the steps of the proposed algorithm in detail where the angle discretization and selection, node orientation, neighbor discovery and ray classification techniques are discussed. Section 5 reports on the performance of the ray classification techniques. Finally, Section 6 concludes the paper.

## 2. Related Work

In general, factoring in the SSP variability in underwater environments yields better estimates of acoustic propagation paths and eventually improves localization accuracy [9]. Since acoustic propagation in an underwater environment tends to refract due to SSP variations, ray tracing models such as Bellhop are employed to find the actual path of underwater acoustic signals [7]. On the other hand, underwater range estimation using time of flight (ToF) is most commonly used in underwater networks, where the range is obtained by correlating the measured ToF and the angle of arrival [10]. However, short distances are assumed among nodes and possible refractions of acoustic signal are ignored. Moreover, we have modeled the acoustic trajectory of convex links using a second order polynomial and proven that for a convex link, a parabola could be used to approximate the actual signal propagation path for ranges less than 3 km [11].

Underwater acoustic waves travel in an inhomogeneous environment with varying speeds [12]. Gao et al. provide a comparison between six different SS estimation algorithms and provide the ranges of temperature, salinity and depths at which each algorithm is applicable [13]. On the other hand, Leroy et al. have proposed a unified equation to estimate the SS across all oceans by introducing the latitude [14]. Due to the varying SS, a different propagation behavior is observed in underwater environments when compared to free space cases. The varying SS has a significant impact on the spreading pattern of an acoustic signal that eventually translates to losses. Taking such fact into consideration, we proposed an efficient method to use broad beams to establish communication links and then proposed narrowing these beams to conserve energy by estimating the transmission and reception angles [8]. However, the method starts by using a broad beam width for neighbor discovery that demands higher power. Moreover, no means are presented to aid nodes in avoiding shadow and overlapping zones by leveraging the capability of the used transducer. Finally, the method cannot suggest new angles when nodes drift or when mobile nodes are used.

In the context of localization, Hasan et al. have studied the effect of increasing the node count on the localization error when a received signal strength indicator (RSSI) is used as a ranging method [15]. The authors attempt to overcome the problems of multipath and fading by increasing the node density. A complete survey for underwater localization schemes is provided in [16], where the tradeoff between various localization methods is highlighted. Emokpae and Younis have proposed a surface-based reflection scheme to perform localization using both line-of-sight and non-line-of-sight links [17]. The authors use an average SS, i.e., a static medium model, to distinguish the different received signals, namely reflected and direct paths. Moreover, Dubrovinskaya et al. exploit the multipath signals between an anchor node and a neighbor to perform ranging by leveraging Time Difference of Arrival (TDoA) among different paths [12]. Since the medium is inhomogeneous and due to the fact that the acoustic wave speed varies along its flight, the authors suggest using an effective SS value derived from the TDoA and the distance that each signal has travelled. However, a rectilinear wave trajectory is assumed, where a single beam follows a straight-line trajectory. Clearly such approximation is not valid for long range communication where acoustic rays tend to bend due to refraction exerted by the medium inhomogeneity. Moreover, a static, pre-defined, model cannot be used to categorize actual propagation paths of acoustic waves that tend to refract with SSP changes. Zhang et al. attempt to use a hybrid localization method where the TDoA and Frequency Difference of Arrival (FDoA) are used to complement each other to predict a more accurate range [18]. They consider the Doppler effect to determine the FDoA and confirm that by factoring in the changes in sound speed across a link, a more accurate result can be obtained.

## 3. System Model and Preliminaries

We assume that each node is equipped with a pressure sensor and an array of transducers to send and receive directional acoustic signals. The pressure sensor is used to estimate the node's depth while the array of transducers enables fine-grained directional transmission. Furthermore, AUNs are assumed to be deployed in deep-water scenario where the separation between the sea surface and

bottom is large. In addition, we assume an obstacle free environment where transmitted rays may change their direction along the depth due to reflections and LEPs but on the average propagate away from the transmitter, i.e., no directional reversal in the horizontal axis. In this section, we discuss the assumed transducer configuration in detail. Then we review the adopted layering and SS estimation methodology. Finally, we discuss how to detect signal reflections and summarize the adopted ray tracing model in a layered environment.

### 3.1. Transducer Configuration and Assumptions

Since we are operating in deep-water environment where separation between the seabed and surface is quite large, links facing more than one reflection are envisioned to lose most of their power and cannot be detected by a receiver. Thus, we only consider in our study the refracted rays and rays facing single reflections, i.e., either from the surface or bottom. Moreover, a transducer is assumed to generate ample power that can overcome experienced underwater attenuation by acoustic signals so that received signal that are refracted or faced a single reflection are decodable. A node is further assumed to be equipped with a transducer that leverages electro-acoustic active elements to generate acoustic pressure waves. Petrauskas shows an example of such a transducer, where the radiation pattern is determined by the shape and size of the used antenna segments [19]. We assume that the transducer can produce a directional beam pattern in the far-field region by means of controlling voltage, phase and amplitude as described by Butler et al. in [20]. Our work assumes that all communication takes place in the far-field region of the radiating element used by the transducer where the pressure waves combine to form a uniform wave-front.

Formally, we define a finite set of beams $(K)$ with cardinality $(n)$ where $n$ is equivalent to the distinct beams that the transducer can generate. Furthermore, we refer to each beam by the center angle of that beam $(k_i)$ and formally write that $k_i \in K$, $i = 1, 2, \ldots, n$. In addition, $\forall k_i \in K$, $\psi_i$ corresponds to the smallest beam width that the transducer can generate at the direction of $k_i$. Clearly $\psi_i$ is dependent on the capability of the transducer in generating directional beams at various directions. Moreover, for $k_i, k_j \in K$ where $i \neq j$ we impose a further restriction that $\psi_i = \psi_j = \psi$ for all possible values of $i$ and $j$, i.e., beam $k_i$ is identical to beam $k_j$ for all values of $i$ and $j$. We then define a spherical transmission envelope across the node, where each beam $k_i$ $\forall i$ is said to have transitioned from the transducer's near field region to the far field. We then regard the node to be at the center of such a spherical transmission envelope. Moreover, we treat the spherical transmission surface as a point source since the distance between each pair in the network is much greater than the radius of such a sphere and the node size. In addition, a node is assumed to be capable of transmitting beams at distinct angles in all 3D directions where $\cup ki$ produces a non-uniform acoustic pressure along the entire spherical envelope surface that is much greater than zero. Moreover, we utilize the smallest beam width, i.e., $\psi$, that a transducer can generate, in order to establish communication links and find the vertical axis. Finally, a beam is visualized as infinite number of rays with infinitesimal angular deviation that span the width of the selected beam.

### 3.2. Underwater Layering and Sound Speed Profiling

The underwater medium is viewed as a layered model based on the observed SSP of the water column. Since the horizontal changes in SS are limited whereas changes in the vertical direction are significant, we use a layered approach to discretize and capture SS changes along the depth [21]. In principal, the SSP of the water column is divided into isogradient layers where each layer has a constant gradient. Since the operational environment is 3D, we represent each layer as a disk with a thickness that is equivalent to the thickness of layer along the *z*-axis. Therefore, the layered model could be visualized as stacked disks, as shown in Figure 3, where the SS gradient in each layer is assumed to be constant. We further assume that nodes obtain the SSP of the water column either by measuring the water column SSP or referencing data collected by field experiments in the same deployment area [22]. Finally, we have proposed a method to determine the SSP from established

communication links where we exploit the fact that the paths of acoustic signals are highly correlated to the SSP of the water column [23]. In the event that the SSP changes, we assume that either such a technique can be leveraged to deduce the changes in the SSP, or new measurements are taken by the node to reflect the latest SSP.

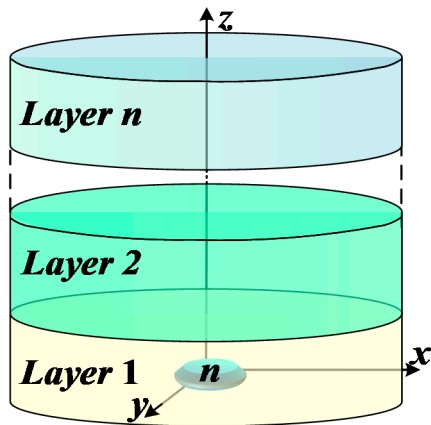

**Figure 3.** 3D layered model with a node $n$ located at the origin of the coordinate system.

### 3.3. Detecting Surface Reflection

Since underwater environments are bounded by the sea surface and bottom, it is inevitable for most of the acoustic transmissions to bounce from either of these two boundaries. The only exceptions, where signals never bounce, are signals forming the Sound Fixing and Ranging (SOFAR) channel where multiple LEPs prohibit the signal from reaching the boundaries. Reflected signals in underwater environments exhibit a phase change that is dependent on the boundary they bounced from. Since the specific acoustic impedance of air-water-interface is roughly 3500 times higher than that of air, a sea-surface reflected signal will exhibit a phase reversal of 180°, i.e., a phase shift of $\pi$. Meanwhile, the phase change of signals reflected from the seabed ($\rho_{bed}$) are given by [24]:

$$\rho_{bed} = -2\tan^{-1}\frac{\sqrt{(\sin\theta_i)^2 - \mu^2}}{v\,\cos\theta_i} \tag{1}$$

where $\theta_i$ is the incident angle on the sea bottom, $\mu$ is the ratio of the water SS to the bottom material SS, and $v$ is the ratio of the density of the sea bottom material to the water density of the layer just above the seabed. Moreover, we assume that nodes are synchronized to a level that enables a pair to identify the phase shift among exchanged beacons. Finally, we assume that signals reflected more than once from any boundary experience high phase distortion and lose most of their power, and consequently cannot be decoded.

### 3.4. Tracing Rays in a Layered Underwater Environment

Ray tracing is referred to the process of predicting the location of acoustic signal at different points and deducing a propagation path. Since sound is characterized as a sequence of pressure waves that propagates in underwater environment, ray tracing attempts to find the path taken by each ray where a ray has an infinitesimal beam width. Moreover, since the beam width of the ray is approaching zero, a ray is not allowed to spread along its propagation path. Therefore, a ray trace is represented using a curve where the slope along the curve is SSP-dependent. When a ray crosses a layer with a linear gradient $g$, it experiences similar refraction conditions at each point within that layer that are governed by the law of refraction (Snell's law) and represented mathematically as:

$$\xi = \frac{\cos\theta_0}{c_0} \tag{2}$$

where $\theta_0$ and $c_0$ are the ray gazing angle and SS observed at some point, respectively. Hence, the propagating ray within a layer will bend with a constant rate and produce a curved path that can be approximated by a circular arc with radius of [25]:

$$R = \frac{1}{g\xi} \tag{3}$$

## 4. Angle Selection and Ray Categorization

Our proposed transmission angle (beam) selection technique aids nodes in conserving energy by minimizing beam overlaps and ensures maximum communication coverage by eliminating shadow zones. The balance of this section presents how to overcome shadow and overlapping zones by exploiting the beamforming capability of the used acoustic transducer to select best angles for establishing communication links. Then we show how nodes can individually map the 3D operational environment into an equivalent 2D one that captures the SSP along the depth. Once the 2D environment with the established links is obtained, nodes begin the process of classifying links based on the experienced propagation path. Finally, nodes divide the underwater environment into distinct regions based on the observed SSP coupled with the ray types. Specifically, each region is obtained by determining the coverage area of distinct beam types.

### 4.1. Angle Discretization and Selection

We consider each transmission angle as a separate communication channel as described in Section 3.1, where the aim is to find channels with the least overlaps and avoid generating shadow zones. Since beam overlapping at large distances is inevitable due to reflections and LEPs, we select beams that avoid both overlapping and shadow regions up to a distance $(d_s)$ where LEP and/or reflections do not occur. Recall from Section 3.1, that nodes can generate identical beams in shape and size and transmit them at distinct directions with a beam width of $\psi$ that is dictated by the transducer capability. Two distinct transmitted beams could (i) overlap, (ii) create a shadow zone or (iii) stage contiguously. We formally define two beams to be overlapping if their center angles deviation is less than $\psi$, i.e., $k_i - k_j < \psi$ for $k_i \neq k_j$. On the other hand, two beams form a shadow zone if $k_i - k_j > \psi$, i.e., the angular deviation between two neighboring beams is higher than $\psi$. A contiguous beam corresponds to $k_i - k_j = \psi$. Leveraging these formal definitions, a node starts the beam selection process where the goal is to select angles that are contiguous to avoid both overlapping beams and shadow zone.

Nodes strive to find the smallest subset $\kappa$ of beams, i.e., $\kappa \subset K$, such that (i) any two adjacent beams in $\kappa$ are contiguous, i.e., has angular deviation of $\psi$, and (ii) the union of all beams in $\kappa$ avoids shadow zones within a distance $d_s$. To find the beams in $\kappa$, nodes define a hypothetical sphere referred in our work as the Transmission Sphere (TS). The radius of such TS is the distance at which any generated acoustic beam $(k_i)$ transitions from the near field region to the far field region of the transducer. Since the assumed transducer enables transmission at distinct directions in all 3D space, nodes can eliminate shadow zones by selecting beams where the union of all beam's footprints onto the TS are capable of completely covering its surface with minimal overlaps. To find the beams constituting the set $\kappa$, nodes maps the TS surface into a grid, where each grid element represents the footprint of a transmitted beam. Fundamentally, nodes divide the TS surface into cells that closely resembles the departing beam's footprint onto the TS. Since a beam is visualized as having a conical shape, the intersection of the TS with the departing beam ideally forms a circular shape that is inscribed onto the TS. However, stacking non-overlapping and tangent circular grid elements onto the TS surface will generate shadow regions since the union of all non-overlapping circles cannot cover the entire sphere surface. Moreover, having overlapping circles inherits the problems of overlapping beams. On the other hand, having irregularly shaped cells significantly complicates the analysis.

Since the choice of the grid element significantly impacts the complexity of beam tracing in an inhomogeneous habitat, we pursue a geodesic grid that strikes a balance between avoiding beam overlapping, eliminating shadow region and faithfully representing the departing beam in shape and size [26]. The simplest and basic geodesic grid is shown in Figure 4a that is obtained by inscribing an icosahedron into the TS surface where each face is represented as a triangle. However, a triangle is quite far from the ideal circular beam-footprints onto the TS and the fact that an icosahedron in its basic form poorly maps the sphere makes it an inappropriate choice. However, subdividing the icosahedron by iteratively bisecting the triangles and projecting them onto the sphere generates a smoother surface approximating the TS as shown in Figure 4b. In the figure, each point on the grid has six edges connecting it to its neighboring points forming a hexagon shape footprint except for 12 points that have five edges that form a pentagon. The pentagon shapes lay on the triangular edges of the basic icosahedron that are referred to as the pentagonal points.

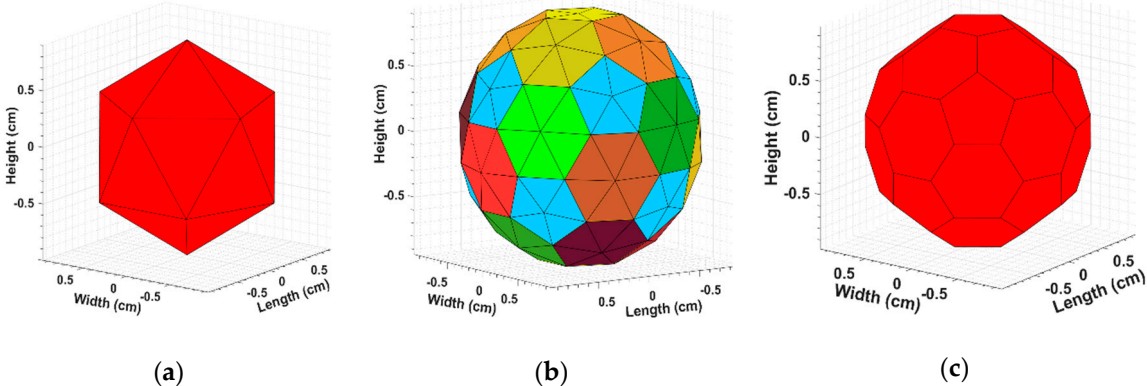

**Figure 4.** Transmission sphere segmentation using a geodesic grid; (**a**) an icosahedron inscribed on a circle, (**b**) subdivided icosahedron by bisecting each triangle and projecting onto the sphere generates pentagons, shown in light blue, and hexagons, and (**c**) a rotated and scaled version of the Voronoi polygon.

Furthermore, subdividing the triangles and projecting them onto the TS surface defines a Delaunay grid that forms finer hexagon and pentagon shapes. We then construct a Voronoi polygon on top of the Delaunay grid by connecting the centroids of neighboring triangles as shown in Figure 4c. Hence, the sphere can be potentially covered with both hexagon and pentagon cells to insure full surface coverage. The triangular subdivision process mentioned above can be repeated until the hexagon and pentagon size is close to the transducer footprint. Once the Voronoi polygons size and shape closely match the footprint, the angle discretization process concludes, and each polygon defines a channel with the transmission angle being the angle connecting the polygon centroid to the transmitting node (center of the TS).

### 4.2. Node Orientation

Since angle of arrival (AoA) based ranging techniques utilizes the angles to obtain the range in a distributed manner, all nodes within an AUN ought to have a sense of orientation relative to the sea surface and bottom. Basically, nodes individually need to infer a depth axis from which they will measure AoA angles. Therefore, after deployment and selecting the feasible angles to avoid shadow and overlapping regions, a node utilizes the natural forces of buoyancy and gravity to roughly orient itself and obtain the positive direction of the vertical axis denoted by *z*-axis [27]. Since using buoyancy and gravity orientation is subject to errors, an error mitigation mechanism is required to accurately determine the vertical direction, i.e., the *z*-axis. In our approach, nodes leverage the selected set of beams $\kappa$ to refine the orientation of a node, as explained next.

Let us assume that node $n_a$ picks $q$ distinct transmission channels where $|\kappa| = q$. Then, beacons are transmitted on every channel such that each beacon has a fixed phase $(\rho_0)$ and is appended by a unique identifier. Once all messages are transmitted, $n_a$ enters a listening phase where it is particularly interested in receiving a reflected version of its transmitted message from the same angle it was transmitted on. Basically, identifying a reflected signal is a two-step process where first $n_a$ compares the identifier extracted from the received reflected signal to that of the transmitted message in the same direction. Since the underwater medium is inhomogeneous and non-symmetrical in general, only reflected signal from the sea surface, sea bottom and nearby obstacles are envisaged to bounce back to the transmitter. Moreover, the surface and bottom reflections will be limited to those beams incorporating a ray that is perpendicular to both the underwater layer boundaries and the bouncing surfaces; this is because any inclination of the ray will be amplified by both refraction and surface reflection and will cause the ray to divert away from the sender, especially for deep water scenarios as shown in Figure 5.

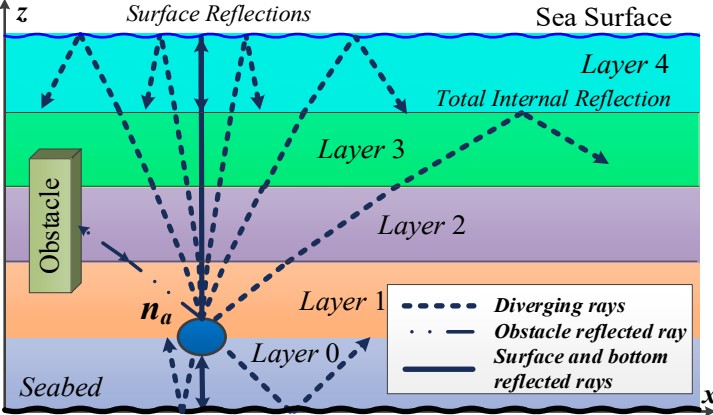

**Figure 5.** Conceptual 2D representation of the layered model with refraction on the edge of each layer. Only rays orthogonal to both underwater layers and sea surface/bottom as well as nearby obstacles will reflect to the node.

When $n_a$ confirms the reception of a reflected message, it calculates the distance to the object $(z_{TOF})$ causing the reflection by factoring in the ToF and the measured SS in its vicinity. Then, $n_a$ computes the phase shift experienced by the received signal $(\rho_r)$ and stores $(z_{TOF}, \theta_T, \rho_r)$ in a candidate list where $\theta_T$ corresponds to the transmission angle of that beam. After accounting for all reflected signals, node $n_a$ utilizes the pressure sensor to estimate the depth $(z_p)$ and looks for the best match within the candidate list where $\rho_T \approx \pi$ and $z_{TOF} \approx z_p$. The best match should also have a direction that is close to the one found by the natural forces of buoyancy and gravity. Once a match is found, $n_a$ regards the center angle of that beam as the initial direction to the surface $(z_i)$, as shown in Figure 6. Note that a transmitted beam is envisioned as a collection of rays that form a conical shape initiating at $n_a$. Moreover, if a match was not found in the candidate list that represents the surface reflection, but a bottom reflection was found, the node utilizes the bottom reflection in a similar fashion to fork a negative and then positive $z$-axis. Furthermore, reflected ray from obstacles will not be able to fulfill all three constraints satisfied by the surface reflection, i.e., having a depth $z_T \approx z_p$, experiencing a phase shift of $\pi$, and having a direction close to one found by using the natural forces of buoyancy and gravity. Finally, if neither bottom nor surface reflections are observed, nodes rely on the already-found axis using the natural forces of buoyancy and gravity. In such a case, errors will be introduced in identifying the location of a LEP as discussed in Section 4.8.

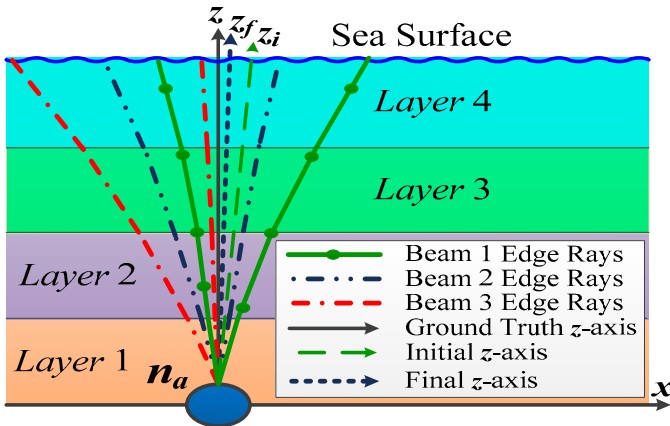

**Figure 6.** Conceptual 2D representation of the z-axis refinement process where bisecting beam 1 containing the surface reflected ray orients towards $z_i$. Then, by transmitting Beams 2 and 3, and detecting their reflections, the node adjusts the direction of the z-axis to $z_f$ that bisects the angle between the furthest edge rays of beams 1 and 2.

The final refinement of the vertical direction is done by exploiting the capability of the transducer to steer a beam, where the underlying assumption is that the steering angle is smaller than $\psi/2$. Since $n_a$ knows that the z-axis lies within the width of the selected beam, it steers the chosen beam towards and away from an arbitrary selected x-axis that is perpendicular to $z_i$ until the reflected version of the transmitted beam is no longer received. The node then notes both final angles from which a reflection was received while steering away and towards the x-axis. Then, the node concludes the direction of the z-axis, also referred as the vertical axis, as the angle bisecting the range of angles it has noted. An example is given in Figure 6 where *beam* 1 represents the beam containing the surface reflected ray and thus selects $z_i$ as its initial candidate to the positive z-axis. Node $n_a$ then transmits *beam 2* and *beam 3* that have a deviation less than $\psi/2$ from $z_i$ and waits for their reflection. Since *beam 3* does not contain the surface reflected ray, its reflection will never reach $n_a$ and will be timed out. Then, bisecting the angle formed by the furthest edges of *beam 1* and *beam 2*, $n_a$ will conclude that the final direction to the surface is $z_f$. Finally, $n_a$ selects an x-axis and y-axis randomly that are perpendicular to each other and to $z_f$.

Once the positive z-direction is finalized, the node searches for another reflection in the candidate list that has an angle deviation of $180 \pm \frac{\psi}{2}$ signifying a bottom reflection. If the entry is found, the node confirms its positive z-axis and forks a negative axis that is 180° from the finalized positive one. Then the node utilizes the bottom reflected signal in order to determine the phase shift that such a signal has experienced ($\overline{\rho_{bed}}$) and store this value for future use. On the other hand, if none of the entries in the candidate list match the bottom reflected signal, the node regards the distance to the bottom as infinity.

### 4.3. Neighbor Discovery

Once oriented, a node announces its presence to neighbors using the selected channels (picked using the procedure in Section 4.1). To do so, a node first represents each beam in the set $\kappa$ using the spherical coordinate system $(r, \theta_i, \phi_i)$, as shown in Figure 7, where $r$ is the radius of TS, $i = 1, 2, \ldots, |\kappa|$, and $\theta$ and $\phi$ are the angles measured from the transmitter's z-axis and x-axis, respectively. Moreover, we will use $n_T, i$ to refer to beam $i$ that is transmitted by node $n_T$. To announce its presence and establish communication links, a "*hello*" message is composed where different headers for each beam $k_i \in \kappa$ are composed that contains: (1) the node depth ($z_T$), (2) the corresponding beam width ($\psi$), (3) the transmission direction $\left(\theta_{n_T,i}, \phi_{n_T,i}\right)$, and (4) the local measured SS ($c_T$). Then for each angle, the corresponding "*hello*" message is transmitted announcing the presence of node $n_T$ to its neighbors and supplying the receiving nodes with data to infer the type of signal, i.e., refracted or reflected.

Upon hearing a discovery message, a receiver $n_R$ notes the values provided by the sender. In addition, node $n_R$ notes the angle at which it received the signal relative to its own local coordinate system $\left(\theta_{n_R,i}, \phi_{n_R,i}\right)$, its measured depth ($z_R$) and the measured SS at the receiver ($c_R$). Such information is linked to each "*hello*" message to enable determining if a ray faced reflection at a later stage. Once all information is collected, node $n_R$ projects the 3D environment into an equivalent 2D one, detailed next.

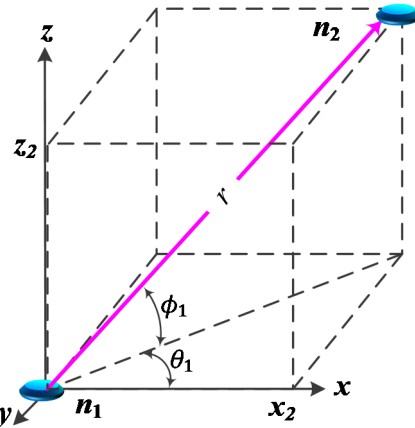

**Figure 7.** Representation of the spherical coordinate system where the angle $\theta$ lies in $xy$ plane, the angle $\phi$ lies in the $yz$ plane and the length of the vector connecting the nodes is denoted by $r$.

*4.4. Defining a Common Coordinate System*

As a result of the node orientation stage discussed in Section 4.2, nodes $n_T$ and $n_R$ obtain a common $z$-axis. Moreover, nodes visualize a beam as an infinite number of rays with an infinitesimal angular deviation that spans the entire beam width. Using such a definition, nodes attempts to identify the "connecting-rays" within a beam. As shown in Figure 8, a connecting-ray is the shortest ray responsible for establishing a link and carrying information between a node pair, i.e., $n_T$ and $n_R$. Recall from Section 3.2 that refraction in the horizontal domain is ignored; thus, if a transmitted ray spans some vertical plane, it can never leave that plane due to refraction. Therefore, it is safe to conclude that a set of transmitted connecting-rays will only span the subspace of a vertical plane ($\Omega$) in their journey from the transmitter to a receiver. Specifically, $\Omega$ must satisfy the following two properties: (1) $\Omega$ passes through both nodes $n_T$ and $n_R$, (2) $\Omega$ incorporates the transmitting node's $z$-axis as a proper subset (or equivalently the $z$-axis of $n_R$ since the orientation stage, discussed in Section 4.2, yields the $z$-axis in the same direction for all nodes). Nodes then selects $\Omega$ and forks an $x$-axis with the following three properties: (1) $x$-axis is perpendicular to the $z$-axis of $n_R$, (2) positive $x$-axis direction gazes at the direction of the received "*hello*" message, and (3) the $x$-axis is a proper subset of $\Omega$. Figure 8 shows examples of $x$-$z$ planes for two connecting-rays ($n_2, n_1$) and ($n_3, n_1$). Finally, the receiver node, $n_R$, regards itself as the origin of the coordinate system it has just formed, i.e., the receiver lies at the intersection of the $z$- and $x$-axis. In the example of Figure 8, $n_1$ will be the origin for the two coordinate systems with respect to $n_2$ and $n_3$.

Once $n_R$ identifies the vertical subspace $\Omega_i$ for each established link $k_i$, it creates a projection of all these subspaces in 2D. Since the underwater environment is modeled as stacked disks, each subspace $\Omega_i$ must have corresponding layer boundaries defined at similar depths. Moreover, due to ignoring refraction in the horizontal directions, a node projects the 3D space into an equivalent 2D by simply superimposing the subspaces $\Omega_i$ formed by different links. Specifically, superimposing is achieved by selecting an $x$-axis as a reference and then rotating each vertical plane $\Omega_i$ along the $z$-axis until all other $x$-axis coincide to the chosen one as shown on the right on Figure 8. In Figure 8, nodes $n_2$ and $n_3$ located at depth $z_2$ and $z_3$ are establishing links with node $n_1$ via the connecting-rays. Since both connecting-rays span distinct subspaces, node $n_1$ preserves the transmitters' depth as well as the behavior of the connecting-rays along the depth by rotating both subspaces along its $z$-axis until both

subspaces overlap. Specifically, the subspace $\Omega_1$ is rotated by an angle of $\theta_1$ while $\Omega_2$ is rotated by an angle of $\theta_2$. Thus, the transmitted signals at different subspaces in 3D are combined into 2D while preserving the acoustic signal behavior along the depth. Nodes utilizes this 2D plane to estimate the range of the connecting-rays and to find the SSP. Such a 2D plane suffices for ranging and for identifying reflected rays, as we show later.

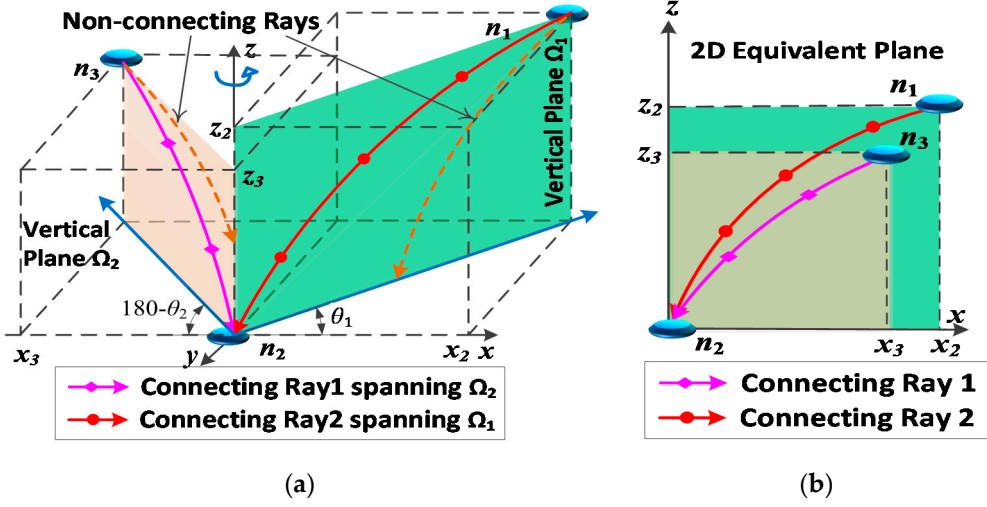

<div align="center">(<b>a</b>)        (<b>b</b>)</div>

**Figure 8.** Conceptual representation of connecting rays where two distinct vertical subspaces are shown with their respective connecting-rays. To map the 3D environment (**a**) into 2D (**b**), we rotate each vertical plane along the *z*-axis until all the vertical planes merger into a single plane referred to as the process of superimposing subspaces.

### 4.5. Ray Types

Transmitted acoustic signals in an underwater environment generally propagate in an inhomogeneous medium that is bounded by the sea surface and bottom. Such bounded inhomogeneous medium often advocates different propagation paths for rays transmitted at different angles and depths. To predict rays suitable for tracking underwater nodes and/or to maintain a communication link among mobile or drifted nodes, it is inadequate to localize the node and know its location only but rather it is important to know the type of ray establishing such a connection, i.e., direct, reflected or refracted with LEPs. Specifically, when separation among a node pair varies due to mobility or drift, different ray types are required to maintain the communication link since each ray can only cover the underwater medium partially. In addition, medium inhomogeneity not only controls refraction of transmitted signals, but it also dictates the number of angles at which the receiver node can be reached. Thus, a node ought to have some means of classifying rays and know the range covered by each ray type to maintain communication with drifted or mobile nodes.

Since the underwater environment is bounded by the sea surface and bottom, it is inevitable for rays to get reflected from these surfaces or refract to the point such that the propagation direction along the vertical axis is inverted. Essentially, reflected rays bounce from the sea surface and/or bottom and may also face refraction along their journey. The point of reflection introduces a LEP point where the signal propagation path is non-differentiable, i.e., a discontinuity in the first derivative is observed. On the other hand, non-reflected rays can be categorized as either rays experiencing a LEP due to medium inhomogeneity or rays that reach the receiver before either a surface reflection or LEP is experienced. Specifically, the former type includes rays that do not bounce from any surface but faced refraction that causes a directional inversion along the *z*-axis and thus generating LEPs along their propagation path. For a refracted ray, the first derivative of the propagation path always generates a continuous signal along the entire path including the LEP point. On the other hand, the latter type is referred to as the direct ray and is defined as the ray with the shortest propagation path between

a communicating pair that face neither a LEP nor reflection from any surface, i.e., faced zero LEPs. A direct ray may still be refracted leading to a curvy propagation path but on the average is either an increasing or decreasing function in $\Omega$. Based on the above definitions, the node starts to classify rays using the measured SS, angle of transmission and the quadrant ($Q$) at which the ray is transmitted and received as discussed in the next section.

### 4.6. Ray Classification and Selection

The proposed ray classification process enables nodes in an AUN to associate established links to the ray types identified in Section 4.5, namely direct, reflected and refracted rays facing LEPs. Since the classification process utilizes the neighbor coordinates in inferring the ray type, nodes go through some ranging and localization algorithm [9,11,18] to obtain a relative map to neighbors. Once localized, nodes apply Algorithm 1 to infer the type of ray based on the approximated propagation path. According to Algorithm 1, node $n_T$ utilizes the angle of departure and the measured SS reading ($c_T$) to infer the SS value at the first possible LEP ($c_{LEP}$). To infer the value of SS at the LEP point, nodes start by assuming that the ray is purely refracted where the steps taken will gauge the validity of such an assumption. Since the gazing angle at the LEP point must be zero, i.e., $\phi_{LEP} = 0$, the law of refraction can be used to obtain the value of $c_{LEP}$ as follows:

$$c_{LEP} = \frac{c_T}{\cos \phi_T} \tag{4}$$

Node $n_T$ considers the SSP of the region from its depth till the depth of the surface facing the transmission angle, and searches for the value of $c_{LEP}$. If $c_{LEP}$ is larger than the maximum value within the SSP of the considered region, node $n_T$ concludes that a reflection is inevitable and falsifies the assumption of having a purely refracted ray. In such a case, node $n_T$ utilizes a straight-line propagation path to estimate the location of the LEP as expressed below:

$$x_{LEP} = x_T + \frac{|z_{LEP} - z_T|}{|\tan \phi_T|} \tag{5}$$

where $z_{LEP}$ in the above equation is either zero or the depth of the seabed depending on the surface the transmission angle is facing. Although the use of the straight-line propagation model does not factor in medium inhomogeneity, it suffices in identifying the type of received ray within a range of 5 km as will be shown in Section 5.2. On the other hand, if $c_{LEP}$ is lower than the maximum value found in the SSP of the considered region, node $n_T$ confirms its assumption of purely refracted ray and extracts the $z$-coordinate of the LEP ($z_{LEP}$). Node $n_T$ then factors its depth ($z_T$) and the corresponding measured value ($c_T$) to compute the SS gradient of the layer enclosing both pairs and the transmitted ray, i.e., $g = (c_T - c_{LEP})/(z_T - z_{LEP})$. Once the gradient is obtained, a node employs Equations (2) and (3) to deduce a radius ($R_{LEP}$). Then, the $x_{LEP}$ for refracted rays facing a differentiable LEP can be found by using simple trigonometry manipulations, as illustrated in Figure 9. Essentially, nodes regard the depth in the interval $[z_T, z_{LEP}]$ as a single layer and find the circular arc that approximates the signal propagation path. Since the transmission angle forms a tangent to the estimated circular propagation path and due to the fact that the tangent is perpendicular to the line connecting the node to the center of the circle, the center of the circle can be uniquely identified. Moreover, since the gazing angle is zero at the LEP point, the line connecting the LEP point and the center of the circular arc is always parallel to the $z$-axis. Thus, $x_{LEP}$ can be mathematically expressed as follows:

$$x_{LEP} = x_T + \left| \frac{c_T(z_{LEP} - z_T)}{(c_{LEP} - c_T)\cos(\phi_T)} \right| \cos\left(90 - |\phi_T|\right) \tag{6}$$

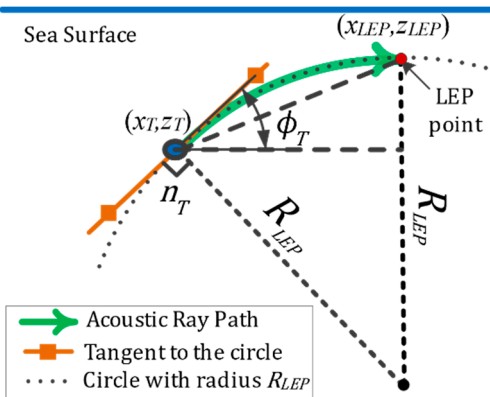

**Figure 9.** Conceptual figure where the maximum horizontal distance for a given transmission angle is presented.

---

**Algorithm 1** Steps to find $x_{LEP}$

---

1:    Calculate $c_{LEP}$ using Equation (4)
2:    Find maximum SS value within SSP ($c_m$)
3:    Extract SSP_Segment = SSP($z_T$, depth of the surface facing the transmission angle)
4:    Find the depth $z_{LEP}$ of the closest SS value within SSP_Segment to $c_{LEP}$
5:    **If** ($c_{LEP} < c_m$)
6:      **If** ($z_{LEP} == 0$) **or** ($z_{LEP} == sebed\ depth$), i.e., differentiable LEP is not possible
7:        Calculate $x_{LEP}$ using Equation (5), i.e., straight-line approximation
8:      **Else**
9:        Calculate gradient $g = (c_T - c_{LEP})/(z_T - z_{LEP})$
10:        **If** $g$ is positive, i.e., differentiable LEP is not possible
11:          Calculate $x_{LEP}$ using Equation (5), i.e. straight-line approximation
12:        **Else**
13:          Calculate $x_{LEP}$ using Equation (6)
14:        **Endif**
15:      **Endif**
16:    **Else**
17:      Calculate $x_{LEP}$ using Equation (5), i.e., straight-line approximation
18:    **Endif**

---

Once the value of $x_{LEP}$ is determined either by using Equation (5) or (6), node $n_T$ compares the coordinate of node $n_R = (x_R, z_R)$ to the LEP coordinate. If the node lies before the first LEP, i.e., $x_R \leq x_{LEP}$, a direct ray is deduced. On the other hand, if $x_R > x_{LEP}$ the node attempts to find the coordinate of the next LEP and repeats the process until the $x$-coordinate of the LEP grows more than the $x_R$. The node then notes the number of LEP required before reaching $x_R$. Once the number of LEPs are determined, nodes utilize the quadrants in which the beam was transmitted and received to classify the ray connecting each pair. The balance of this section details such a process.

Once the location of the LEP point is determined, nodes consider the quadrants of transmission and reception as well as the depth of the LEP point to determine the ray type. Specifically, since we are working in an obstacle free deep-water scenario, a transmitted signal will never face a directional reversal along the $x$-axis and thus a ray that is transmitted within the range of the first quadrant ($Q1$) will always be received either in the 2nd or 3rd quadrant ($Q2$, $Q3$) as shown in Figure 1b. For rays transmitted within $Q1$ and being received within $Q3$ while the transmitter is located at a deeper point than the receiver, it is safe to assume that an even number of LEPs will be experienced. Fundamentally, if the number of LEP is zero, a node concludes that a direct ray has been received where the signal was intercepted by the receiver before a LEP point is experienced. Hence, nodes rule out experiencing reflection or LEPs since the receiver is located at a point where neither is possible. Moreover, since an

odd number of LEPs cannot satisfy the condition on the receiving quadrant, experiencing any type of single LEP is ruled out. On the other hand, if rays are transmitted from *Q*1 and received in *Q*2 while knowing that the transmitter is at higher depth, a receiver concludes that an odd number of LEPs is experienced thus ruling out direct ray paths. Table 1 summarizes the number of LEPs observed when transmitting at different angles and depth configurations. The table emphasizes the fact that direct rays are only observed when transmission and reception angles adhere to *Q*1 and *Q*3 while the transmitter is located at a greater depth or when rays fall in *Q*4, *Q*3 and the transmitter is at a shallower location. In both cases reflections is also ruled out. Moreover, a direct ray cannot be observed for all other cases. Reflection, on the other hand, is observed only when the number of LEPs is greater than zero. Similarly, refracted rays with multiple LEPs may be observed depending on SSP conditions. Specifically, experiencing a differentiable LEP requires the SS along the propagation path to be on the average an increasing function, i.e., the ray must be traversing from a lower SS value to a higher one. Therefore, the gradient of the SSP controls the horizontal distance before an acoustic signal experiences a LEP, where generally lower SS gradients result in LEP point closer to the transmitter. Thus, to distinguish between the non-differentiable LEP from the differentiable ones, i.e., reflected from refracted rays experiencing a LEP, a node considers the number of experienced LEPs along the propagation path and their depths. If the depth of a LEP point corresponds to the sea surface or bottom, nodes conclude that the ray faced a reflection. On the other hand, if the LEP depth is lower than the surface the ray is gazing at, refraction causing a differentiable LEP is deduced.

**Table 1.** Ray classification based on node depths and quadrants of the transmitted and reception.

| Quadrant | | Depths | Number of LEPs $n \in \mathbb{Z}^*$, i.e., Set of Non-Negative Integers |
|:---:|:---:|:---:|:---:|
| **Tx** | **Rx** | | |
| 2,3 | 1,4 | x [1] | Impossible |
| 1 | 3 | $z_T \geq z_R$ | $2n$ |
| | | $z_T < z_R$ | $2n + 2$ |
| 1 | 2 | x | $2n + 1$ |
| 4 | 3 | x | $2n + 1$ |
| 4 | 2 | $z_T > z_R$ | $2n + 2$ |
| | | $z_T \leq z_R$ | $2n$ |

[1] x represents a don't-care value.

In summary, using the technique presented in this section, nodes calculate the number of LEPs faced in the propagation path and determine the type of ray that establishes a communication link. Moreover, by using such a technique, nodes do not have to go through the entire Bellhop algorithm [28] but rather uses simple and less complex equations to determine the expected number of LEPs along the propagation path. Once the rays are classified into their appropriate type, a node determines the usable range of each ray type, as discussed in the next section.

*4.7. Underwater Regions Based on Ray Types*

The ray classification technique discussed in Section 4.5 determines the type of ray given an established communication link, yet, it cannot provide the range beyond which one type of ray cannot be used by a transmitter to reach a receiver. To determine the range of each ray type, nodes leverage the SSP and the possible transmission angles to determine the location of LEPs. Since the transmission angle is bounded in the open interval $\phi_T \in (-90, 90)$, nodes first estimate the location of LEPs for all transmitted signals. Specifically, using Equation (4) the range of angles ($\nu_{LEP}$) at which $c_{LEP} \leq c_m$ is first calculated. For each angle within $\nu_{LEP}$, the node leverages Equation (6) and notes the location of the LEPs. Utilizing the maximum angular value in $\nu_{LEP}$, a node applies Equation (5) to determine the location of a non-differentiable LEP. As shown in Figure 10, connecting all LEPs noted by node $n_T$

yields a contour $(\lambda_{LEP})$ that can be leveraged to identify four different zones. The first zone is labeled in Figure 10 as Region 1 where only direct and surface reflected rays can establish a communication link. In the second zone, Region 2, only reflected and rays facing a single LEP can reach the receiver, i.e., direct ray connections are ruled out. For the third zone, labeled as Region 3, the direct ray is forced to spread out to the extent that it drops below the receiver sensitivity especially for long range communication. In particular, a beam bounded by Ray 5 and Ray 6 does not spread much before reaching $\lambda_{LEP}$. However, as soon as the beam approaches the contour, a rapid stretch is exerted onto the beam's wave-front by the SSP as shown by the size of the lines representing the wave-front in Figure 10. Considering the spreading losses only, the law of energy conservation states that the energy must be evenly distributed along the curve representing the wave-front of a beam. Furthermore, since a line approaching $\lambda_{LEP}$ from the left and right significantly vary in size, the beam's energy must spread evenly across such lines. As a consequence, beams transmitted with a finite energy must spread rapidly and hence the acoustic energy at the receiver transducer located after $\lambda_{LEP}$ will be significantly lowered and becomes undetectable. Thus, for Region 3, neither direct rays nor rays facing a differentiable LEP can reach and only reflected rays can cover such a region. The final zone is the white space shown in Figure 10; such a zone is not reachable by $n_T$ using any angle, i.e., a permanent shadow zone.

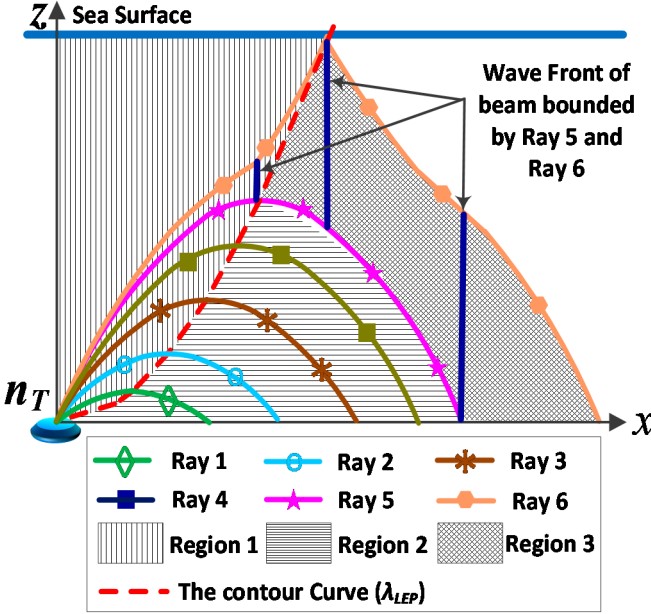

**Figure 10.** Demonstration of different regions generated in underwater environment where SSP of the water column dictates the boundary of each one.

In summary, the SSP of the water column dictates the propagation path of an acoustic signal. Therefore, whenever the SSP changes nodes need to obtain a new set of reading to reflect the most recent SSP. Based on the observed refraction and reflection of rays, the underwater environment can be categorized into four regions based on the type of ray observed in each one. A region in which none of the rays can reach is referred to as the permanent shadow zone. To identify the location of the permanent shadow zone, a node must find the ray path of two distinct rays. The first is identified by an angle that generates the furthest underwater differentiable LEP. The second ray is obtained by identifying the ray with the furthest non-differentiable LEP achievable within the SSP. Therefore, the permanent shadow zone is only created in two SSP conditions where the first requires the SSP between a node and the reflection surface to be on the average an increasing function and thus producing LEPs along the entire depth of the SSP column. The second condition is when the SS gradient increases in value and then shifts and becomes either constant or on the average a decreasing function indicating that differentiable LEPs will be possible at the depth at which the shift occurs.

The second region is where the direct ray resides and is always bounded by the transmitter node's *x*-axis and *z*-axis as well as the contour $\lambda_{LEP}$. For the third region, direct rays cannot reach while rays facing a differentiable LEP can. The final region is located between all three regions mentioned above where only surface reflected rays can reach this region. Moreover, reflected rays can be found in all regions except the permanent shadow zone. Finally, not all regions exist in each underwater environment where distinct SSPs may have fewer regions when comparing to others.

*4.8. The Effect of Angle and Depth Errors*

The categorization algorithm presented in Section 4.6, utilizes the angle of transmission to estimate the ray type. Since a node starts by orienting itself, i.e., finds the vertical axis, and then extrapolates the horizontal axis, any error in determining the vertical axis will affect the accuracy of the transmission angle. If we denote the error in obtaining the vertical axis by "*e*", the error in calculating $c_{LEP}$ from Equation (4) is given by:

$$c_{LEP} = \frac{c_T}{\cos(\phi_T + e)} \tag{7}$$

Using simple trigonometric function, the denominator can be rewritten as $\cos(\phi_T + e) = \cos(\phi_T)\cos(e) - \sin(\phi_T)\sin(e)$. Moreover, since the value of *e* is assumed to be small, we can approximate $\cos(e) = 1$ and $\sin(e) = 0$ and write the denominator as $\cos(\phi_T) + e\sin(\phi_T)$. Finally, since the value of $\sin(\phi_T)$ cannot grow more than one, the worst error can be expressed as follows:

$$c_{LEP} = \frac{c_T}{\cos(\phi_T) + e} \tag{8}$$

Thus, the error in the angle can be viewed as an additive error to the cosine of that angle. In Section 5.2, we provide a plot that shows the error trend when a gaussian error is superimposed on the actual angle value.

The error introduced when the depth is inaccurately measured has a very different pattern. Basically, the error in the depth directly affects the position of $x_{LEP}$. Since our algorithm utilizes Equation (5) or (6) to determine the distance to the LEP, we discuss the error in each equation separately. Through Equation (5), a node concludes that a differentiable LEP is not possible and hence resorting to the straight-line model is necessary. If we replace the value of $z_T$ with the erroneous depth, i.e., $z_{Te} = z_T + e$ where "*e*" is the error observed in the depth, the error term can be extracted and is found to be additive, where the value of the additive error is given by $\left(e \times \tan^{-1}\phi_T\right)$. Thus, the error in this situation will be scaled by the value suggested by the inverse tangent function and is dependent on the angle of transmission. On the other hand, Equation (6) assumes that the ray is refracting and hence a different error pattern is determined. Since only the fraction shown in Equation (6) contains the value of $z_T$, we focus our attention on the fraction since all other parameters will not be affected. By substituting the value of $z_{Te} = z_T + e$ into $z_T$, one can show that the error term generated will be an additive error with a value given by:

$$\text{Depth} - \text{caused Error} = \frac{e}{\left(\frac{c_{LEP}}{c_T} - 1\right)\cos(\phi_T)} \tag{9}$$

Moreover, since the temperature in underwater environment is limited between 30 °C and −1.7 °C, the achievable SS values in practical oceanic environments are limited within the interval $c_{int} = [1400, 1580]$ (m/s) [29]. Using such limits in Equation (4) yields a maximum angle of transmission of 27.6° to have a LEP. Furthermore, the ratio $(c_{LEP}/c_T)$ is always greater than one since $c_{LEP}$ must be greater than $c_T$ to achieve a LEP. Using the values within $c_{int}$ we can obtain a maximum value for the ratio in actual underwater conditions and show that it cannot grow beyond 1.13. Coupling all the information above, a maximum bound on the depth-caused error is given by $8.77 \times e$ in actual sea water conditions. Thus, when determining the value of $x_{LEP}$ for a differentiable LEP and if an error

equivalent to "*e*" is experienced in determining the depth, the position of $x_{LEP}$ will face a maximum error of $8.77 \times e$.

## 5. Performance Validation

The beam selection mechanism and the related steps of orientation, node discovery and projecting the 3D environment into 2D, are validated by comparing the proposed ray classification technique with the types suggested by the Bellhop ray trace. The balance of this section describes the simulation environment and reports the results.

### 5.1. Simulation Setup and Node Deployment

To mimic actual underwater conditions, we have used an actual SSP from the 2013 World Ocean Database (WOD) [30]. Specifically, we employ the Leroy model [14] on the eXpendable Bathythermograph (XBT) data was collected by the ship NOAAS OKEANOS Explorer on June 5th, 2009 and May 25, 2010 to obtain the two different SSPs shown in Figure 11. Since the temperature readings obtained by the ship for M-SSP1 terminate at 1617.5 m while the maximum measured seabed depth is 3249 m, the SSP after 1617.5 m is treated as depth-dependent only. The selection of M-SSP1 is due to its close resemblance to the Munk SSP profile which is deemed as an idealized SSP and is widely used in underwater simulation [31]. Furthermore, M-SSP2 inhibits a different SS gradient along the depth and provides an underwater environment that is quite far from the Munk profile and hence acts as a good testing baseline.

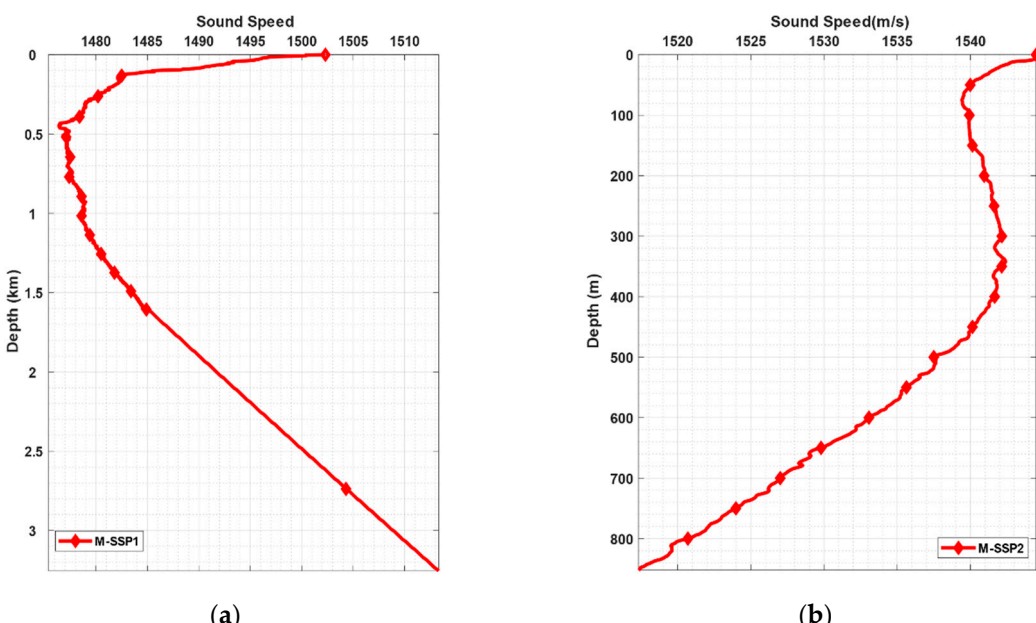

(**a**)                                        (**b**)

**Figure 11.** The (Sound Speed Profiles) (SSP) extracted from World Oceanic Database (WOD) database with part (**a**) showing the first SSP (labeled M-SSP1) measured at coordinate of N40° 31′ 24.179″ W127° 35′ 52.27″, and (**b**) signifies the second SSP (labeled M-SSP2) measured at N20° 7′ 18.0624″ W 175° 56′ 17.3436″.

In our simulation, we first select one of the shown SSPs in Figure 11 and then deploy nodes randomly within a region spanning the depth of the SSP and up to a horizontal distance of 5 km. We then utilize such a plane as the equivalent vertical plane chosen in the 3D environment. We then run our ray categorization algorithm and identify direct ray, rays experiencing a single LEP and those undergoing multiple LEPs. We then select the direct rays in validating the performance of the proposed ray classification technique.

*5.2. Performance Results*

To validate our approach, we have deployed nodes in different configurations and within both M-SSP1 and M-SSP2. To fully test the proposed algorithm's capability, we generated within M-SSP1 and M-SSP2 more than 1600 rays and used the proposed ray classification technique to determine the ray type. When applying our approach, 96% of the rays are classified correctly while only 4% were misclassified. We have chosen four different configurations shown in Figure 12 as examples to fully comprehend the reasons behind the observed 4% error. The figure displays the Bellhop eigen rays generated for node #3 and the location of its neighbors. In the configuration shown in Figure 12b our approach manages to classify all rays correctly except for two rays. The first ray terminates at node #7, where the depth of the node is 1.4 m, i.e., is located very close to the sea surface. In this case the value of $c_{LEP}$ is higher than the maximum SSP value; thus, a reflection was assumed and the value of $x_{LEP}$ was obtained using a straight-line approximation. The location of the $x_{LEP}$ was found to be 18.3 m before the location of node $x_R$ in an actual horizontal range of 800 m, i.e., an error of 2.28% in calculating the location of the LEP. Thus, the algorithm assumes that the ray was received immediately after it was reflected from the sea surface and categorized the direct ray into a ray facing a single nondifferentiable LEP. Similar conclusion can be made for ray terminating at node #17 where the ray was classified falsely as having three LEPs while it experienced a single LEP. A closer look at the location of the second LEP reveals that it was estimated to be 5.5 m before the actual location of the node. Since the quadrants of transmission and reception mandates and odd number of LEPs, the nodes opted to choose a three LEP propagation path. Moreover, in both cases, the proximity of the node is extremely close to the location of the LEP and given the actual range value the error is classified as insignificant. Such errors can be filtered out by adding some error cushion to the actual LEP location where if a node is within a marginal proximity of the LEP, it ignores the last LEP. When classifying the rays using the configuration used in Figure 12c, the direct ray terminating at node #20 was erroneously categorized as having a single LEP. Since the calculated value of $c_{LEP}$ was larger than the observed SS in the water column, the algorithm opted to use a straight-line approximation and found the first LEP point 78 m before the node location. Moreover, node #20 is located at a horizontal distance of 4938 m, i.e., at the extent of the simulation area. At such a large distance, the straight-line model managed to retain an absolute error of 78 m and hence is deemed appropriate for estimating reflected ray paths to an extent of 5 km.

In part (e) and (f) of Figure 12, M-SSP2 is utilized to show different refraction patterns than those used in the previous examples. Since most of the transmission angles in part (e) are close to the horizontal axis and due to the fact that the SSP is not symmetrical, a new environment is introduced to validate the proposed algorithm. We observe that the ray classification algorithm produces error only in three cases and in the remaining 40 rays shown in parts (e) and (f) were classified correctly. For the 3 cases, similar findings are observed as those found in part (b) and (c). Thus, the four different configurations shown in Figure 12 demonstrate the capability of our ray classification technique in identifying the ray type for node located at ranges less than 5 km.

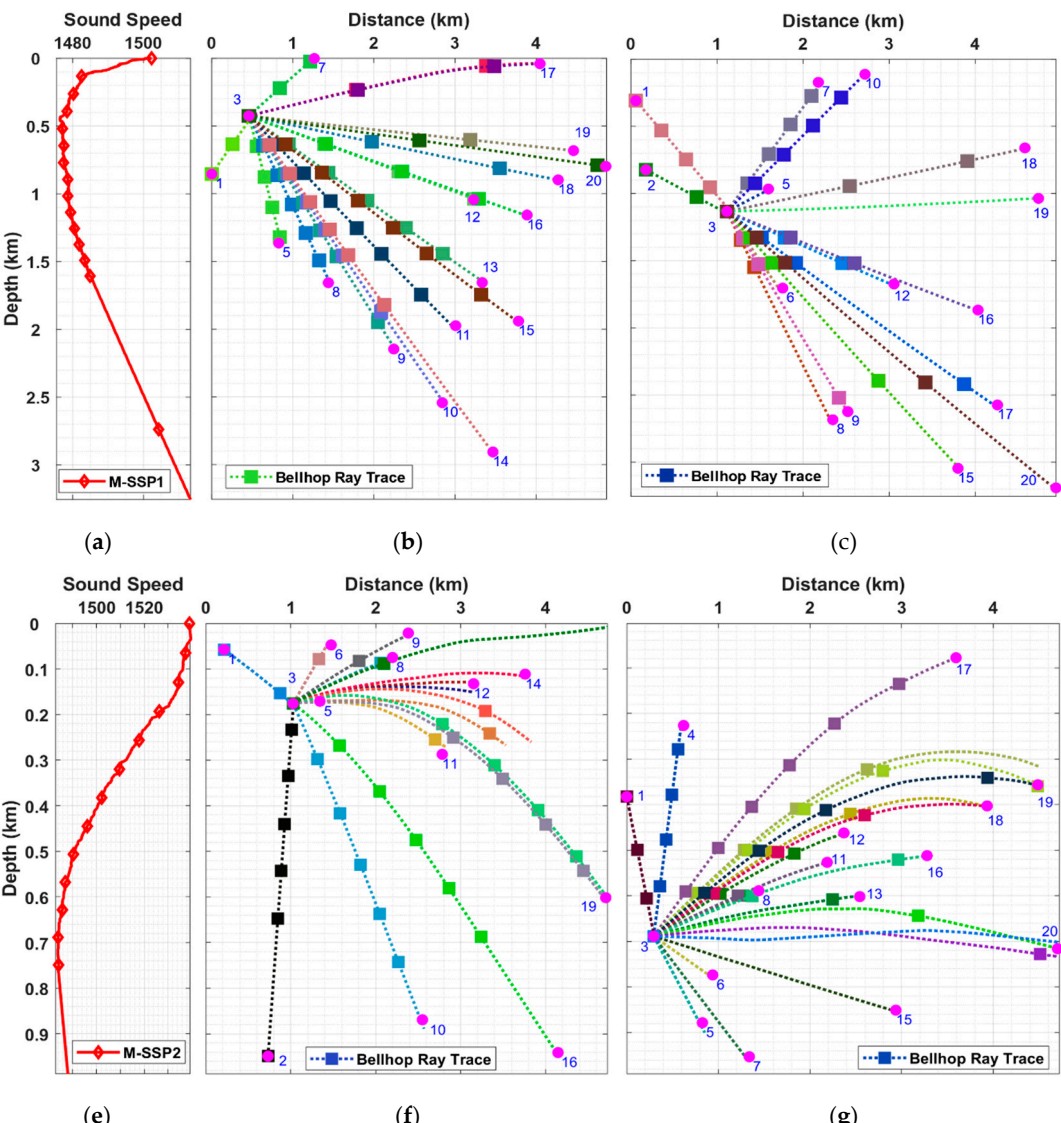

**Figure 12.** Propagation path of selected acoustic signals shown in different deployment configurations. Part (**a**,**d**) show the SSP for the figures in the same row while parts (**b**–**e**) show the node distribution for each deployment configuration and the corresponding refracted rays only among pairs.

Figure 13 shows the percentage of ray misclassification when selecting an erroneous angle (labeled "Angle Error"). Specifically, we inject an additive Gaussian error with zero mean and variable standard deviation ($\sigma$) values to the selected angle and observe how many signals are misclassified. From the figure, we can see that when the transmission angle is error free, less than 5% of the rays are wrongfully classified. As we introduce a Gaussian error $\mathcal{N}(0, \sigma)$ we find that on the average, the error grows almost linearly with the increasing value of $\sigma$. Thus, even when the node orientation step produces an error, the effect of such an error on the ray classification technique is bounded and does not grow to a level that the technique is not usable. Furthermore, since the expected error in detecting the orientation described in Section 4.2 often results in a maximum error equivalent to the smallest beam width achievable by the transducer and its capability to steer such a beam, the orientation error is envisioned to be small.

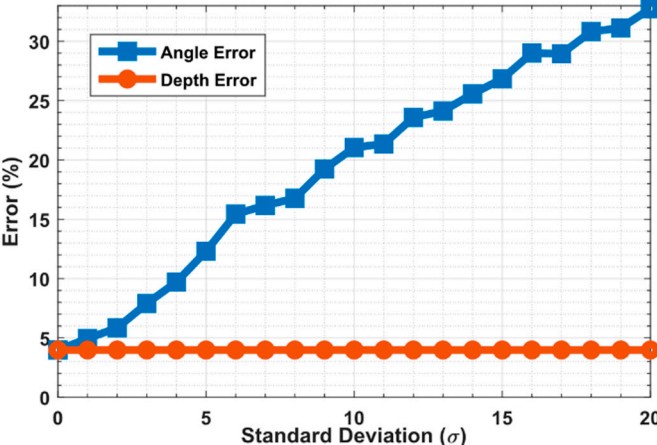

**Figure 13.** The classification error trend when the angle of transmission and depth are subjected to an additive gaussian error with zero mean and standard deviation depicted on the *x*-axis.

The error resulting from an erroneous depth (labeled "Depth Error") estimation is also simulated and presented in Figure 13 where we have added a Gaussian error $\mathcal{N}(0,\sigma)$ to the actual depth of the transmitter node. Figure 13 shows that the effect of gaussian error up to $\sigma = 20$ m is negligible. Such observation is supported by the fact that in a randomly deployed topology, the chances of having two nodes close to a bouncing surface, communicating at transmission angles less than five degrees and use a straight-line propagation model is quite slim. Moreover, for rays with differentiable LEPs we need to determine the error produced by a pressure sensor. Specifically, since the depth is obtained by a pressure sensor, the accuracy of the depth is dependent on the hardware used by a node. The depth sensor "OceanDEPTH" that is manufactured by "OceanTools" produces an error of 0.25% [32]. Thus, the value of the error for depth up to three kilometers for such a sensor will be $e = 7.5$ m. We utilize such an error value to plot the experienced error in $x_{LEP}$ and show the result in Figure 14. Specifically, when the straight-line approximation is utilized the error is represented in Figure 14a while the maximum error bound for the differentiable LEP is presented in Figure 14b.

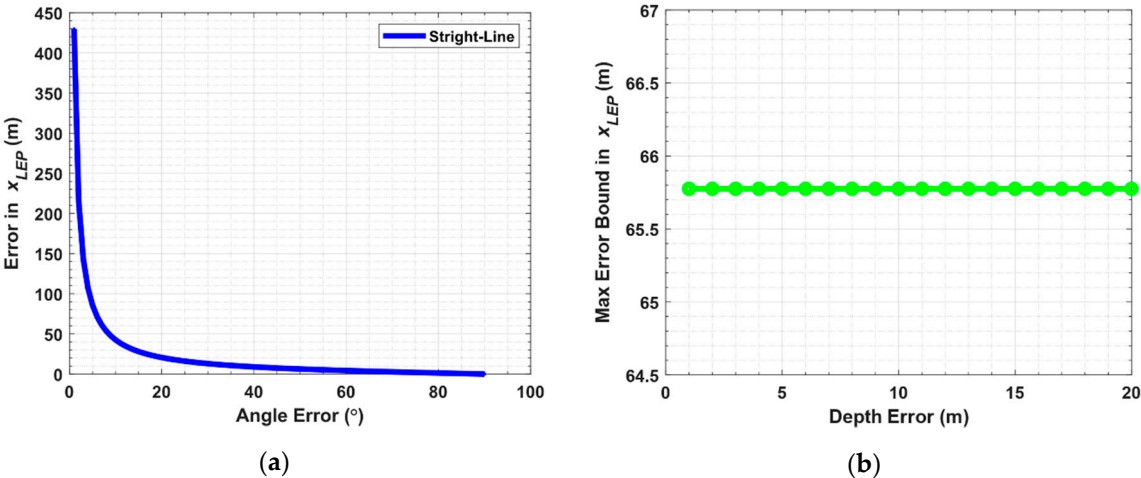

(**a**)                    (**b**)

**Figure 14.** The error observed in the $x_{LEP}$ where (**a**) is the error when Equation (5) used while (**b**) shows the maximum error bound when Equation (6) is used. Both figures assume that the error by the pressure sensor is $e = 7.5$ m.

In summary, our algorithm is quite robust and correctly classifies rays even with errors introduced in the angle and depth. The algorithm is more sensitive to errors in the transmission angle while the effect of a depth error is generally insignificant. For the depth-error cases, the corresponding classification errors for the straight-line model has been found to be dependent on the transmission

angle; such effect diminishes for higher angles. The depth-caused differentiable LEP error, on the other hand, is bounded by a maximum cap. Finally, when utilizing a pressure sensor with different accuracy, the generated error pattern will only be a linear translation along the *y*-axis of the cases shown in Figure 14.

## 6. Conclusions

The paper presents methods that enables underwater nodes to establish communication links using a directional underwater transducer. Our approach leverages a geodesic grid to identify channels that ensure minimal overlap while maximizing coverage. We show that by using the geodesic grid, a node avoids creating a shadow zone and conserves energy by choosing contiguous beams. We then utilize the selected beams to aid nodes across the AUN in obtaining a common depth axis in a distributive manner. To simplify the analysis, a projection scheme is presented that maps the 3D underwater environment into a 2D one that preserves the SSP structure and is deemed suitable for studying the inhomogeneous effects on propagating underwater acoustic signals. The paper also presents a lightweight underwater ray classification algorithm that identifies the ray types based on their propagation path. Specifically, the classification algorithm categorizes rays into direct, reflected and a refracted ray that change the vertical direction due to SS changes along their path. The classification algorithm performance was validated in a simulated environment using actual SSP measurements. For more than 1600 randomly deployed node pairs within a range of 5 km, 96% of the rays were classified correctly. Finally, to aid nodes in tracking underwater targets or locate drifted nodes and reestablishing communication with them, we compute the location of contours between regions to aid node in deciding which type of rays is most suitable to communicate within each region. Using the different regions, it was found that only the surface reflected rays can cover all regions except the permanent shadow zone for which communication cannot be established unless the location of the node changes. Moreover, the permanent shadow zone is only observed when refraction forces acoustic signals to change the vertical direction, i.e., face a LEP. Finally, regions where direct rays and rays facing a directional reversal before they reach a bouncing surface are also identified. By exploiting these regions, underwater nodes are more capable of tracking and reestablishing communication with mobile or drifted nodes. Furthermore, we discuss the effect of angle and depth errors on the categorization algorithm.

**Author Contributions:** A.A. and M.Y. contributed in conceptualization of this article. Moreover, the proposed methodology, software and simulation, validation, formal analysis, investigation, data curation, visualization and original draft preparation was conducted by A.A. M.Y. helped in refining the algorithms, reviewing the results, and editing the final manuscript.

**Funding:** This work is supported in part by the National Science Foundation, USA, contract #0000010465. Moreover, Akram Ahmed is also associated with and supported by King Fahd University of Petroleum and Minerals, Kingdom of Saudi Arabia.

**Conflicts of Interest:** The authors declare no conflict of interest.

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
