# Peer review of "Acoustic Beam Characterization and Selection for Optimized Underwater Communication"

_applsci, doi:10.3390/app9132740_

Round 1

Reviewer 1 Report

This paper presents the beam selection method that enables long-range communication among underwater noses by using realistic sound speed profile. The idea of ray beam classification is interesting and sufficiently simulated in the 3D underwater environment. The following points should be considered as minor revision.

(1)  pp. 2 line 62

Figure 1(b) has been explained before Figure 1(a). Authors had better consider the order of explaining Figure 1(a) and Figure 1(b).

(2)  pp. 14 Table 1

In the number of LEPs. “n=0,2,4,...” is difficult to understand. Authors had better another notation such as “2n”, “2n+1”, “0 or 2n”. In the depth, the condition of “Z_T = Z_R” is missing.

Author Response

Kindly, find the response to the comments in the attachment where we have prepared a new manuscript ready for submission with all the comments addressed.

Reviewer 2 Report

The article is well written however some minor changes are required to improve the quality of publication. 

1- Two more keywords should be added.

2- Please check the alignment of equation 5.

3- Figures, Table and Algorithm should be centralized..

4- Reference [33] and [34], make the hyperlinks available.

Author Response

(The authors gave the same response as above.)

Reviewer 3 Report

· When you include one acronym, write the first letter of the definition in capital letters. When you add an acronym, you need include the definition before of it allways. You don't include the definition of AUN, RSSI, XBT. You need include the acronym WOOD in line 713 because you use it in the title of figure 10.

· Which is for you a "very lo frequency"? Specify with values or range in line 43.

· In my opinion, is wrong citing in this way: "In [x]...". Please, read more papers and learn that cites are in the end of the sentence that refer to the cite.

· After number include space between the number and the units. Many times you put "Xm" and not "X m".

· Why is not mentioned the first of 1b figure?

· Review the first of 1b figure, I think that has units wrong. Is it Sound speed in m/s, right? not Range in meters. I suggest you plot the range between 1400 and 1500 m/s, not more, to notice the variation in more dtail.

· Why is nr2 is not mentioned in the text?

· Is not better modify the order of figures a and b in figure 1? Because you mention before the figure 1b in the text.

· The correct form to cite figures, equations or tables in the text is using italic format.

· Why you can not put the figure with algorithm 1 in the center and write the text under it?

· Excessive bibliography for a paper. I suggest you only put bibliography from papers, and please add the DOI in each one if it's possible.

· Are you sure that in the line 173 is "latitude" the correct word and not "depth"?

· Clarify the DToA concept. Which is the receiver to calculate it? Between consecutively receivers or you use one like a reference?

· v isnot defined in the equation 1.

· If you includes axis in figure 4 is for include units in the figure.

· If you talk about noise, please add the SNR value that you are expecting in each case.

Some modifications/corrections (suggestions):
· Line 61-63: "To illustrate, let us consider the example in Figure 1(b), where node ??1 lies on the path of the ray transmitted by ?? at an angle ?? = 12°" -> "To illustrate, let us consider the example in Figure 1(b), where first receiver node (??1) lies on the path of the ray transmitted by transmitter receiver node (??) at an angle of transmission (?? of 12°)"

· Line 67: "as illustrated in the Figure 1"

· Add some scheme to clarify theta_t? An scheme to clarify the angles variables.

· Line 112: "in the figure" -> Which figure?

· Use the acronym ToF not TOF

· Use the acronym DToA not TDoA

· Line 318: Transmission Sphere (TS)

· Line 687: "x- and z-axis" -> "x and z axis" or "x-axis and z-axis"

About the work:
· 5 km of propagation is a realistic value for you? Do you have some reference for it?

· Do you consider that's sea current can be modify the acoustic ray path too?

· Do you have a real application for this work?

· Which is the precission of pressure sensors that you are expecting? How it can be affect your results?

· Do you consider the attenuation of the signal?

· Wich type of emitters/receivers is considering on the work?

Author Response

Kindly, find the response to the comments in the attachment where we have prepared a new manuscript ready for submission with all the comments addressed. Please note that the line numbers in the attached response are from the updated manuscript and not the previous submitted one. However, the website does not allow to upload a new manuscript.

Round 2

Reviewer 3 Report

Thanks for your reviews. In special, thanks a lot for the comment 21, I have learned the difference about DToA and TDoA.

Good work.